# Representation of women at American Psychiatric Association annual meetings over 10 years (between 2009 and 2019)

**Sabrina Sebbane**[1☯], **Sophie Bailly**[1☯], **Wayne-Corentin Lambert**[2], **Stéphane Sanchez**[3], **Coraline Hingray**[4,5], **Wissam El-Hage**[1,6]*

**1** Centre Régional de Psychotraumatologie CVL, CHRU de Tours, Tours, France, **2** Department of Hemostasis, CHRU de Tours, Tours, France, **3** Pôle IMEP, Unité Recherche Clinique et de recherche en soins, CH de Troyes, Troyes, France, **4** Centre Psychothérapique de Nancy, Pôle Hospitalo-Universitaire de Psychiatrie d'Adultes du Grand Nancy, Laxou, France, **5** Université de Lorraine, CNRS, CRAN, UMR 7039, Nancy, France, **6** UMR 1253, iBrain, Université de Tours, Inserm, Tours, France

☯ These authors contributed equally to this work.

* wissam.elhage@univ-tours.fr

**Data Availability Statement:** All relevant data are within the manuscript and its Supporting information files.

## Abstract

### Objective

Sex disparity is a major societal issue. The aim of this paper was to describe changes in the representation of women among speakers of the American Psychiatric Association (APA) annual meeting over 10 years, between 2009 and 2019 and to compare them to changes in the proportion of women among American psychiatrists.

### Methods

Data were collected from the programs of the APA annual meetings of 2009 and 2019, and from the Association of American Medical Colleges. Descriptive and comparative statistical analyses were performed.

### Results

There were 1,138 distinct speakers at the 2009 conference and 1,784 at the 2019 conference. The number of distinct female speakers increased from 413 (36.3%) to 813 (45.6%). The proportion of female speakers at the meetings was almost equivalent to the proportion of women in the American psychiatrists' workforce. The number of female chairs increased from 158 (39.6%) to 322 (46.4%). There were 38 female speakers in child and adolescent psychiatry in 2009 (51.4% of 74 speakers) and 74 in 2019 (51.0% of 155 speakers).

### Conclusions

The representation of women at the APA annual meetings increased between 2009 and 2019. At the same time, the growth in the percentage of women in the American psychiatrists' workforce was slower. The APA appears to promote female representation during its annual meetings.

**Funding:** The authors received no specific funding for this work.

**Competing interests:** CH reports personal fees from EISAI, Janssen, Lundbeck, Otsuka and UCB. WEH reports personal fees from Air Liquide, EISAI, Janssen, Lundbeck, Otsuka, UCB, Roche and Chugai. This does not alter our adherence to PLOS ONE policies on sharing data and materials.

## Introduction

As we live in an era promoting sex equity in multiple ways, this question is also extensively studied in healthcare, in particular for healthcare providers [1] and in research [2]. Since the first woman to officially study in an American medical school in 1847 [3], the rates of female physicians have increased markedly, including in psychiatry. Indeed, approximately 40% of active psychiatrists and 50% of active child and adolescent psychiatrists in the United States (US) were women in 2017 [4]. In academic psychiatry, women are now more represented as faculty, but are still underrepresented in leadership roles such as department chair [5].

Annual meetings of scientific societies are major events for physicians, and speaking during these meetings is a good way to promote their work and increase their professional visibility. Studying the rate of women speaking during these meetings could be an interesting way to evaluate their representation and visibility in their field. Also, studying sex disparity could in itself improve sex equity [6]. Previous studies have shown an underrepresentation of women at academic meetings [7–17]. The gap between the percentage of women in the workforce and the percentage of women among speakers, however, seems to decrease with time [7, 9, 15, 18]. Several medical specialties have studied female representation at their annual meetings. For instance, studies showed that sex equity among speakers was achieved in 2015 at the microbiology meeting [6]. In contrast, annual neurosurgery meetings failed to reach sex equity among speakers [19] as in hip and knee surgeon annual meeting [16]. In psychiatry, we found only two studies, which demonstrated that female representation at the French and Australian annual meetings improved over the years [20, 21]. While the proportion of female speakers at the Victoria Branch Conference evolved from one to four in seven between 2013 and 2014, women were still underrepresented at the French annual meeting in 2018 [20, 21].

The American Psychiatric Association (APA) is the world's most prominent psychiatric organization [22]. It openly promotes equity [23] and has published a position statement indicating its intent to increase the representation of Women in Psychiatry (WIP) in leadership roles [24]. Silver et al. [25] examined sex disparity in medical society leadership between 2008 and 2017. They showed that the APA was one of the few medical societies that had achieved sex equity for leadership [25]. Larson et al. [7] showed that WIP were underrepresented among speakers at the APA meeting between 2013 and 2017. However, their study analyzed female representation only among keynote and plenary speakers and invited lecturers [7]. No study to date has analyzed changes in sex disparity among all the APA meeting speakers over an extended period of time.

The aim of this paper was to describe changes in the representation of women among speakers at APA meetings between 2009 and 2019. Another objective was to describe changes in the various roles, the different topics and the different sessions presented in the programs. Finally, we wanted to compare female representation at the APA meetings to female representation in the US psychiatric workforce.

## Methods

This retrospective, cross sectional study evaluated the evolution of the representation of female speakers at the 2019 APA annual meeting compared to the 2009 APA annual meeting. This work also analyzed sex distribution for each role, session and topic described in the meetings. Finally, female speakers representation at the APA meetings was compared to female representation in the US psychiatric workforce.

## Study sample

The programs of the APA annual meetings of 2009 and 2019 were analyzed to accurately collect, for each speaker, four main variables: *i)* Sex; *ii)* Role: chairs, presenters, lecturers, directors, faculty and discussants; *iii)* Session: general sessions, courses, presidential sessions, workshops, master courses and special sessions; *iv)* Topic of the session: addiction psychiatry, child and adolescent psychiatry, consultation-liaison psychiatry, diversity and health disparity, forensic psychiatry, geriatric psychiatry, residents, fellows, and medical students.

## Variables

We collected the roles for each speaker. There were some dissimilarities in types of roles between 2009 and 2019 programs. For comparison purposes, we gathered some types of roles under the same terminology (Table 1). The following terminology was used to classify the

**Table 1. Terminology matching between 2009 and 2019.**

| 2009 | 2019 | Used terminology |
|---|---|---|
| **Roles** | | |
| Chair, Vice-chair | Chair, Host | Chair |
| Director | Director | Director |
| Faculty | Faculty | Faculty |
| Discussant | Discussant | Discussant |
| Participant, Presenters, Pro/con-side, Moderator, Panelist, Not specified | Presenter, Pro-con, Moderator, Not Specified | Presenter |
| **Sessions** | | |
| Lecture | Presidential session | Presidential session |
| Master courses | Master courses | Master courses |
| Mindgames, Special Event | Special session | Special session |
| Workshop | Learning Lab, Media Session | Workshop |
| Symposium, Advances in . . . series, Case conferences, Forums, Scientific and Clinical report session, Medical updates | General session | Symposium |
| Courses | Courses | Courses |
| **Topics** | | |
| Addiction psychiatry, Alcohol and drug related disorders, Eating disorder | Addiction psychiatry, NIDA research track | Addiction psychiatry |
| Attention spectrum disorder, Child and adolescent psychiatry and disorders | Child and adolescent psychiatry | Child and adolescent psychiatry |
| AIDS and HIV Related disorders, Pain management, Sleep disorders | Consultation-liaison psychiatry | Consultation-liaison psychiatry |
| Cross-cultural and minority issues, Ethics and human rights, Gender issues, Lesbian gay bisexual transgender issues, Religion, spirituality and psychiatry, Social and community psychiatry, Stigma advocacy | Diversity and health disparity | Diversity and health disparity |
| Psychiatric education, Resident and medical students concerns | Resident, fellows and medical students | Resident, fellows and medical students |
| Forensic psychiatry | Forensic psychiatry | Forensic psychiatry |
| Geriatric psychiatry | Geriatric psychiatry | Geriatric psychiatry |
| Other* | Unspecified | Other topics |

* Sessions from the "other" topic from 2009 were classified one by one with the most relevant 2019 topic according to their name, if no 2019 topics matched, they were classified as "other topics".

types of roles: chair, director, faculty, discussant and presenter. We distinguished presenters of special sessions from the overall presenters as these speakers appeared to be in the spotlight.

Similarly, we collected speakers sex information from the different types of sessions in the programs. For comparison purposes, we also pooled some types of sessions under the same terminology (Table 1). The following terminology was used to classify the types of session: presidential session, master course, special session, workshop, symposium and course.

The main topic of each session was mentioned in the 2019 program. As some of the topics mentioned in 2009 did not match those in 2019 and were more complex, the authors decided after collegiate discussion to assign one of the 2019 topics to a part of the 2009 sessions. Sessions with no topic mentioned or no topic corresponding to the list were labeled as "other topics". The following terminology was used to classify the topics: addiction psychiatry, child and adolescent psychiatry, consultation-liaison psychiatry, diversity and health disparity, forensic psychiatry, geriatric psychiatry, residents, fellows, and medical students. Some sessions mentioned two different topics.

The sex of the speakers was identified using their first and last names, combined with an Internet search, which was necessary to confirm cases for which direct identification was not possible. Sex identification was performed by one author, then verified by a second one to avoid wrong sex assignment. Finally, as some speakers spoke in several sessions, each speech was counted and defined.

National demographic data were obtained from the website of the Association of American Medical Colleges and the Accreditation Council for Graduate Medical School. We collected data for the 2010–2019 active WIP and child and adolescent psychiatry in the US as 2009 data was not available. The percentage of female residents and fellows in psychiatry and child and adolescent psychiatry in the US in 2009 and 2019 was collected.

## Statistics

We first performed a descriptive analysis. We calculated the percentage of female speakers in the 2009 and 2019 APA annual meetings. We calculated the total percentage of female speakers for each year and then for each type of session, role and topic.

For each year, we performed a comparative analysis to determine whether the percentage of women speakers was different from 2009 compared to 2019, overall and for each category.

Then, we performed a multivariable analysis by logistic regression, evaluating the probability of female participation according to role, type of session, topic and APA year (2019 *vs* 2009). The variables to be included in the logistic regression model were selected to reflect the main differences in congress structure between 2009 and 2019. The analysis was conducted using R version 4.0.2 (www.R-project.org).

## Results

### General data

We describe the change in the percentage of female speakers at the 2009 and 2019 APA annual meetings, according to the inclusion criteria described above. We identified 1,138 and 1,784 distinct speakers at the 2009 and 2019 APA meetings, respectively. The number of female speakers increased from 413 (36.3%) to 813 (45.6%), and was almost equivalent to the proportion of active female psychiatrists in the US. Indeed, the percentage of active female psychiatrists in the US was of 37.1% in 2010 and 42.9% in 2019. The proportion of female residents and fellows in psychiatry, however, decreased from 51% to 48.9% between 2009–2010 and 2019–2020. This suggests that the percentage of female speakers at the APA annual meetings was slightly higher than the percentage of active female psychiatrists during the approximately

same decade. Females intervened in 522 (31.4%) and 926 sessions (40.6%) in 2009 and 2019 respectively, while males intervened in 1,139 (68.6%) and 1,355 (59.4%) sessions. The number of interventions per speaker decreased for both genders: from 1.26 to 1.14 interventions per speaker for women and from 1.57 to 1.40 interventions per speaker for men.

## Roles

Here we describe the representation of female speakers among chairs, directors, faculty, discussants and presenters. The number of female chairs increased from 158 (39.6%) to 322 (46.4%) from 2009 to 2019. There were 286 (36.9%) female presenters in 2009, and 721 (46.5%) in 2019. Although the role of director was less frequent in 2019, a higher proportion of women exerted it that year: the number of female directors changed from 24 (20.7%) to 12 (25.5%): The number of female faculty increased from 47 (34.8%) to 48 (42.9%). There were 6 (27.3%) female discussants in 2009, and 36 (40.9%) a decade later (Fig 1A and 1B). The role was not defined for one female speaker in 2009 and one male speaker in 2019.

## Topics

Here we describe the representation of female speakers for selected topics found in the 2009 and 2019 APA conferences: addiction psychiatry; child and adolescent psychiatry; consultation-liaison psychiatry; diversity and health disparity; forensic psychiatry; geriatric psychiatry; residents, fellows, and medical students. Each of the 3,942 interventions were assigned to one (96.3%) or two (3.7%) main topics. The percentage of female speakers in child and adolescent psychiatry sessions remained stable: 38 women (51.4%) in 2009 and 79 women (51.0%) in 2019. The number of female speakers in consultation-liaison psychiatry sessions increased from 44 (34.9%) to 45 (44.1%). The percentage of female speakers at the APA diversity and health disparity sessions increased from 104 (44.4%) to 125 (55.1%). Likewise, the proportion of female speakers in forensic psychiatry was higher in 2019 (77 women; 42.5%) than in 2009 (34 women; 29.1%). The number of female speakers in geriatric psychiatry increased from 15 (39.5%) to 28 (51.9%). The number of female speakers in "residents, fellows, and medical students" sessions increased from 63 to 164. However, the proportion of women in these sessions remained almost stable (52.1% and 56.7% in 2009 and 2019 respectively). However, the number of women decreased in addiction psychiatry, from 53 (39.8%) in 2009 to 43 (30.7%) in 2019. For other topics, the number of female speakers increased from 186 (29.9%) to 645 (44.1%) (Fig 2A and 2B).

## Sessions

Here we describe the representation of female speakers among general sessions, courses, presidential sessions, workshops, master courses and special sessions. The number of female speakers during presidential sessions changed from 17 (35.4%) to 71 (41.0%). The number of female speakers during master courses decreased from 5 (35.7%) to 4 (19.0%). The percentage of female speakers during special sessions remained similar: 6 (42.9%) *vs* 10 (41.7%). Considering both genders, there were 433 speakers in workshops in 2009 and 83 in 2019. The proportion of female speakers during workshops remained stable: 197 (45.5%) in 2009 vs 46 (55.4%) in 2019. The percentage of female speakers during symposia was higher in 2019 (936 female speakers: 46.2% of speakers in this type of session) compared to 2009 (231 female speakers: 33.1% of speakers in this type of session). The proportion of female speakers in courses was higher in 2019 than in 2009 (65 or 39.7% *vs* 52 or 27.4%) (Fig 3A and 3B). The type of session was not defined in four cases in 2009 and in 36 cases in 2019, of which 20 were sessions presented by female participants.

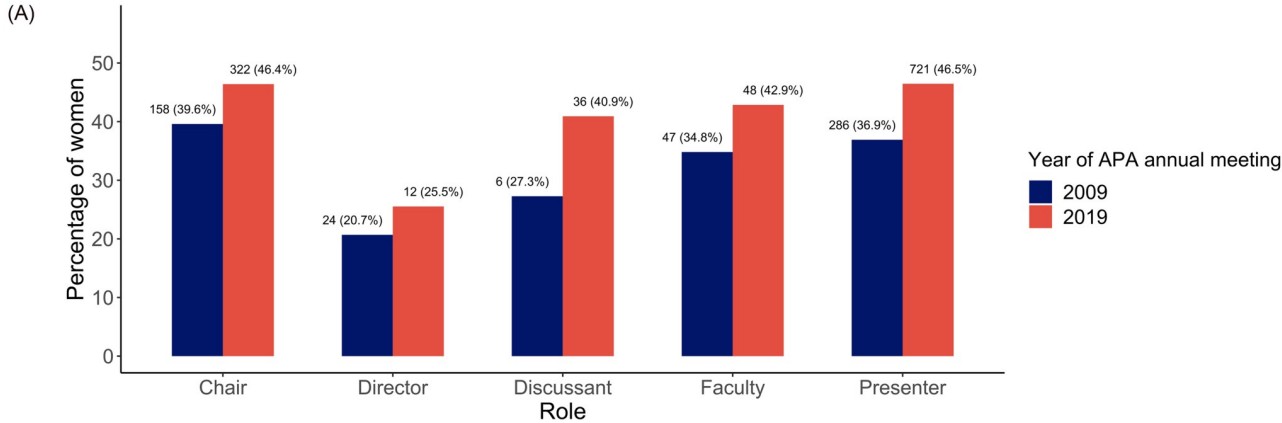

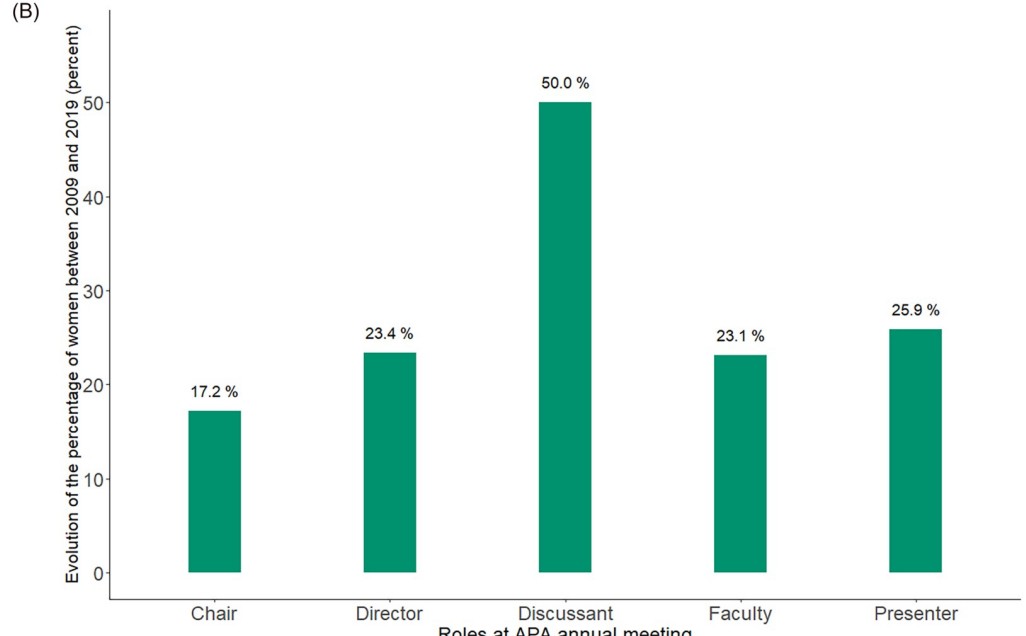

**Fig 1.** A. Rate of women per role, 2009 versus 2019. B. Change in rate of women per role.

### Multivariable analysis

Results of the multivariable analysis are presented in Table 2. The probability of female participation increased between 2009 and 2019, with an Odds Ratio of 1.580 (95%CI 1.351–1.850, p < .001). The Area Under the Curve (AUC) was 0.603. No calibration problems were found, the Hosmer-Lemeshow Goodness-of-fit test was not significant (p = 0.19).

### Discussion

Overall, WIP were more represented in the 2019 APA conference, and the multivariable analysis showed that this increase was independent from the differences in the repartition of topics, roles, or type of sessions observed in 2009 and 2019. We also collected demographic data concerning the American psychiatry workforce. Among the total number of speakers, 45% were

(A)

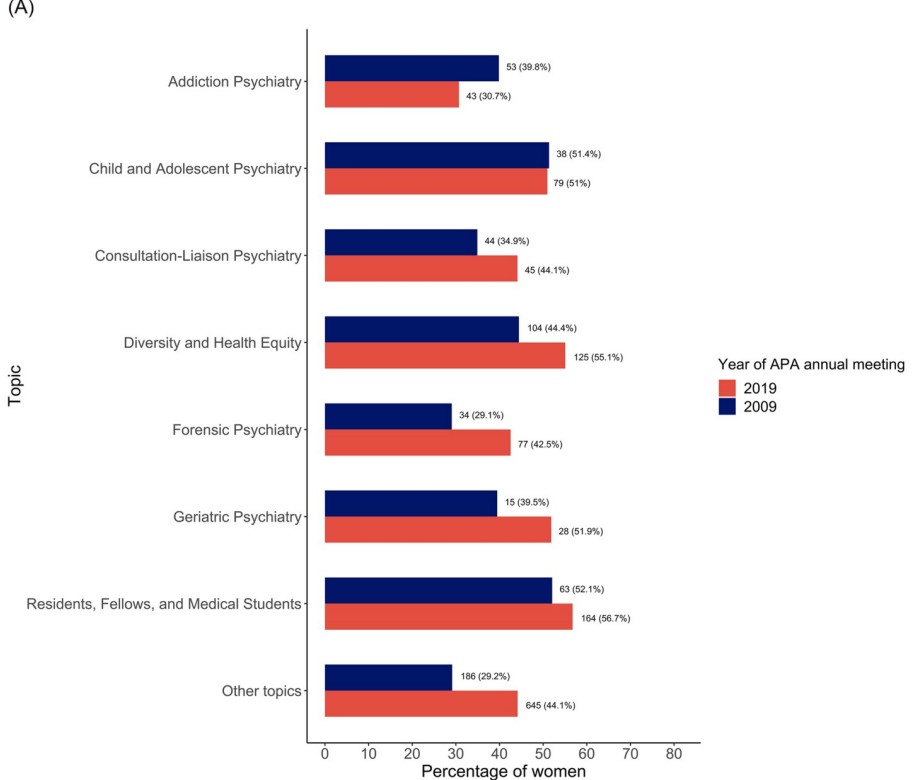

(B)

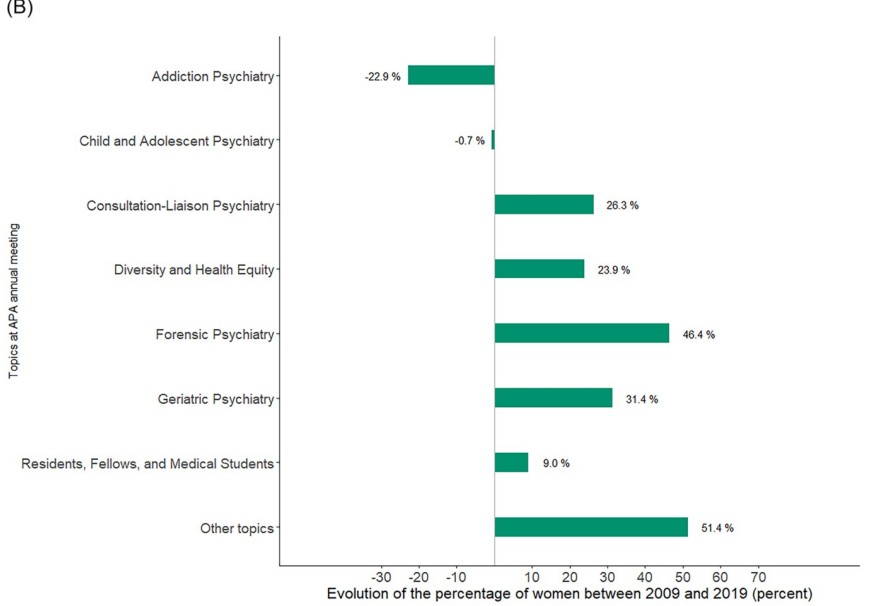

**Fig 2.** A. Rate of women per topic, 2009 *vs* 2019. B. Change in rate of women per topic.

(A)

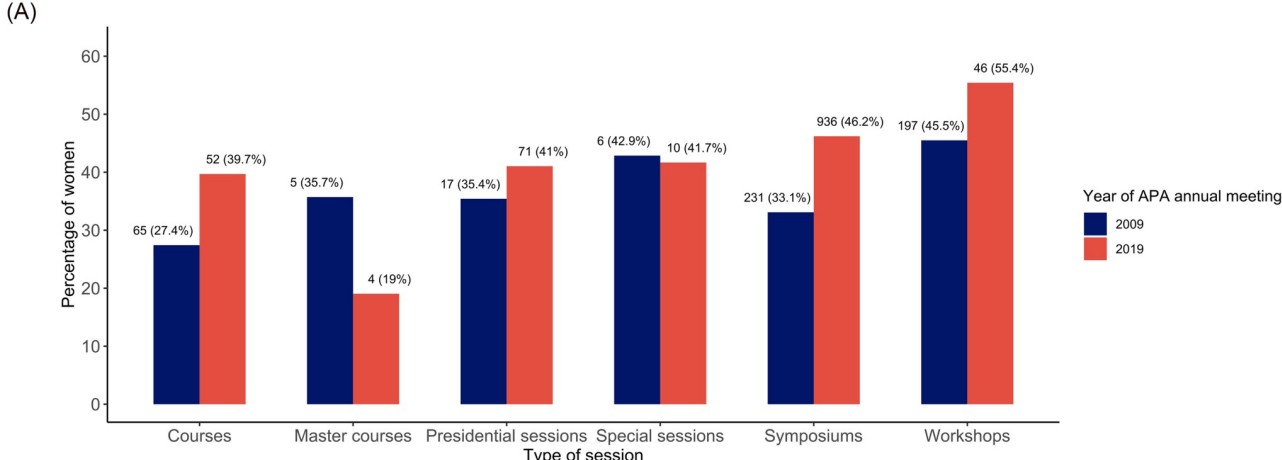

(B)

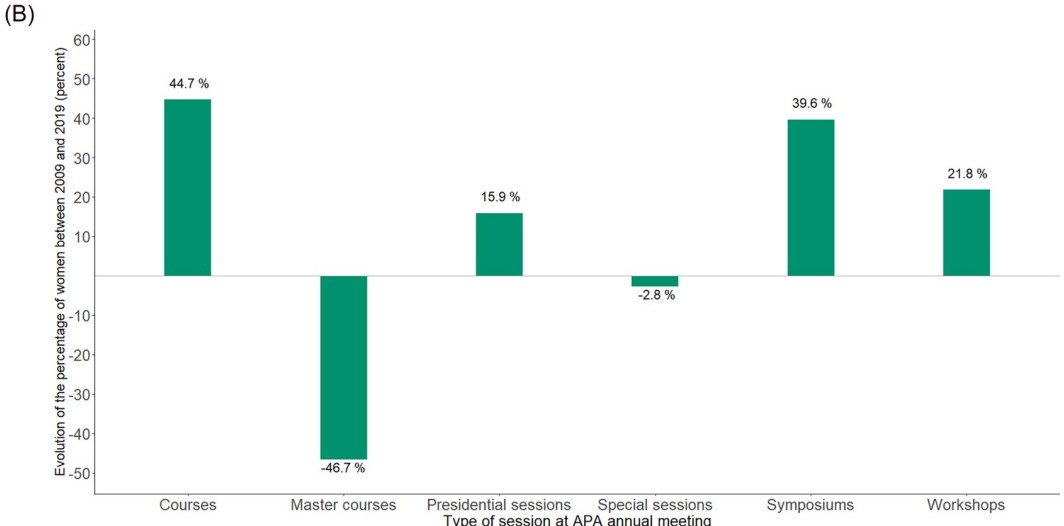

**Fig 3.** A. Rate of women per type of session, 2009 *vs* 2019. B. Change in rate of women per type of session.

women in 2019 versus 37% in 2009, reflecting an increase in the representation of women at the APA annual meeting over the past ten years. The representation of female speakers was higher in almost all the different types of sessions, roles and topics. The representation of female chairs increased, which could suggest that women are taking more leadership positions at the meetings now than ten years ago. Finally, female presenters were more represented in 2019. Presenters constituted the largest category among the different roles. This underlines the fact that women were more present in 2019 than in 2009.

Women were significantly more present in diversity and health disparity sessions in 2019. Consequently, they constituted the majority of speakers in these sessions in 2019. Indeed, WIP seem to be more interested in disparity matters as they may feel more directly concerned. Interestingly, the proportion of female speakers increased significantly in forensic psychiatry. In contrast, female representation in child and adolescent psychiatry decreased, even if slightly. At the same time, female representation in the child and adolescent psychiatry workforce rose in the US (2010–2019). These data also show that women appeared to be overrepresented in child and adolescent psychiatry at the 2009 APA meeting. Nevertheless, the representation of women in addiction psychiatry decreased between 2009 and 2019.

**Table 2. Multivariable analysis by logistic regression modelling the probability of female participation according to APA session.**

| Variable | Odds Ratio | Lower 95CI% | Upper 95CI% | P-value |
|---|---|---|---|---|
| **Year 2019 (ref = 2009)** | **1.58** | **1,351** | **1,85** | **< .001** |
| **Role** | | | | **0.01** |
| Presenter | 1 (reference) | | | |
| Chair | 1.019 | 0.878 | 1.182 | |
| Director | 1.017 | 0.282 | 4.003 | |
| Discussant | 0.839 | 0.557 | 1.251 | |
| Faculty | 2.177 | 0.641 | 8.153 | |
| **Session** | | | | **0.006** |
| Symposium | 1 (reference) | | | |
| Courses | 0.446 | 0.117 | 1.546 | |
| Master courses | 0.288 | 0.063 | 1.173 | |
| Presidential session | 0.864 | 0.648 | 1.147 | |
| Special session | 0.775 | 0.360 | 1.597 | |
| **Workshop** | **1.447** | **1.163** | **1.799** | |
| **Topic** | | | | **< .001** |
| Child and Adolescent Psychiatry | 1 (reference) | | | |
| **Addiction Psychiatry** | **0.486** | **0.334** | **0.706** | |
| **Consultation-Liaison Psychiatry** | **0.613** | **0.412** | **0.908** | |
| Diversity and Health Disparity | 0.888 | 0.636 | 1.237 | |
| **Forensic Psychiatry** | **0.499** | **0.344** | **0.720** | |
| Geriatric Psychiatry | 0.753 | 0.428 | 1.319 | |
| **Other topics** | **0.589** | **0.442** | **0.783** | |
| Residents, Fellows, and Medical Students | 0.967 | 0.684 | 1.365 | |

Area Under the Curve (AUC): 0.603. Hosmer-Lemeshow Goodness-of-fit test: 0.19.

The results showed that female representation at the APA meetings has increased over the past decade. Simultaneously, female representation in honorific sessions did not increase. Indeed, in presidential and special sessions, female representation remained stable between 2009 and 2019. Female representation in master courses fell dramatically during this decade. However, these results can be adjusted by the low number of speakers in these categories. Nevertheless, the representation of women in honorific sessions seems to reflect the lower representation of women in academic medicine.

## Literature review

To our knowledge, this is the first study to precisely evaluate the proportion of women among APA annual meeting speakers. The same work has been conducted for other specialties. Sleeman et al. [13] showed that more women were speaking at palliative care conferences. This was concordant with the fact that the palliative care workforce counted more women than men. Nevertheless, they also found that fewer women were invited as plenary speakers, one of the most honorific roles [13]. In surgery conferences, sex equity seems harder to achieve, as some sessions include an "all-male panel" [11], and changes in representation of women among speakers over time appear quite low [19]. These facts should be considered carefully as women are less present in the surgery workforce compared to the medical workforce [11, 19]. At critical care conferences, female speakers are underrepresented when compared to the percentage of women in the critical care workforce [14]. In emergency medicine, Carley et al. [8]

showed that women speak less frequently and give shorter speeches than men during meetings. In Hip and Knee Surgery meetings, Cohen-Rosenblum et al. showed no evolution of women representation between 2012 and 2019 [16].

In contrast with our work, Pierron et al. [20] found an underrepresentation of female speakers during the French psychiatry meetings (2009–2018) despite a slow growth. Like in our paper, women were also more highly represented in child and adolescent psychiatry sessions. They underlined a low representation of women in chairs, similar to our result on the lower representation of women in APA honorific roles such as special presenter and director [20]. In Australia, sex equity was achieved at the Victorian Branch Conference one year after the program committee became aware of the gender gap [21]. These differences in evolution and representation observed between the American, French and Australian psychiatry meetings could be explained by the cultural particularities of each country. Arora et al. also showed a difference between regions (US, Canada, Australia, UK and Europe) considering women representation in meetings [15]. Overall, each program committee policy could also play a major part in sex disparity among speakers.

## Hypothesis

This work shows the positive change in the representation of women speakers at the APA annual meetings between 2009 and 2019. One hypothesis to explain this favorable evolution is the APA's policy on disparity and diversity. First, the APA is one of the few medical societies respecting sex disparity in the choice of their presidents [25]. This could have impacted the selection of their speakers at the annual meetings. Second, the APA 2009 and 2019 program committees consisted of 50% women and 50% men. Sex parity in the committee program could promote sex parity in the choice of speakers [26]. The APA is also an active partner organizer of the Women's Wellness Through Equity and Leadership (WEL) Program, that promotes female career development. These positive results can be explained by the APA's political will and commitment to the fight against discrimination.

Although the APA promotes sex parity, part of the APA annual meetings did not respect sex equity in the choice of speakers. Women were underrepresented in master courses but also in honorific roles. One hypothesis to explain this underrepresentation could be that fewer women were invited to hold these positions. This might be due to a reduced visibility of women in the academic field stemming from lower opportunity to present their work as invited speakers. A study evaluating the gender disparity in the authorship of the Canadian Cardiovascular Society (CCS) guidelines over two decades showed no change in women authors percentage [27]. Similarly, Rai et al. evaluated the inclusion of women in the American College of Cardiology/American Heart Association guideline writing committees overs 15 years. They showed a persistent gender disparity over the years even though the percentage of women increased. In both studies [28], the authors concluded that further efforts are required to include more women in leaderships roles. Breaking that circle might increase female visibility in academic medicine and make them more likely to be invited as speakers [29]. Another explanation might be that, while WIP made up to 42% of the psychiatry workforce in 2019, they are still underrepresented in academic medicine. Indeed, only 22% of the department chairs in psychiatry were women in 2018 [5]. The lack of WIP in leadership positions might be a reflection of what is called the glass ceiling, which represents the invisible and artificial barriers that stop women from gaining access to senior leadership roles, despite their high representation in the field [30, 31]. Another reason could come from WIP themselves. Indeed, in evolutionary biology symposia, women were found to turn down invitations more than men. Explanations could be low self-promotion (less desire to put themselves forward), more

obstacles related to managing children, a greater reluctance to speak in public or lower perception of their ability and success [29].

## Strengths and limitations

This is the first time that sex disparity has been assessed in a major international psychiatric meeting as large as the APA meetings. As the APA meeting gathers a large number of speakers, the results obtained possess a high statistical power. This high statistical power allowed us to interpret the results observed. One of the strengths of our study is that it did not just assess changes in the proportion of female speakers at the APA meeting. Our work took into consideration the type of sessions, the type of roles and the topics in which the speaker was participating. This analysis allowed us to pinpoint the specific areas where representation of women increased whereas some others still need improvement. Another strength is that it did not use an automatic tool to identify the sex of each speaker. Indeed, a systematic google search of the name of the speaker was performed. In addition, sex identification was performed by two authors independently. Therefore, the risk of wrong sex assignment was quite low.

There are, however, a few limitations in this study. First, the APA annual meeting brings together speakers from the US and other countries. Consequently, a good part of the speakers at the APA annual meeting were probably not American. The demographics collected for comparison, however, included only the US workforce. This is due to the lack of international data on the percentage of female psychiatrists. Another limitation is that some of the speakers were not psychiatrists. Speakers could be physicians from other specialties, but also residents, psychologists, researchers, philosophers, or lawyers. This makes the comparison to the psychiatrist workforce less pertinent. Nevertheless, psychiatrists constituted the majority of speakers, lessening the significance of this bias. Another limitation is that the data were obtained from the annual meeting programs. They did not take into account last-minute changes or cancellations. This could have led to incorrect data collection. Still, changes in programs do not happen frequently and represent only a small part of the program. Consequently, the corresponding bias can be considered minor. Another source of bias in our analysis could have come from the method used to homogenize the difference in categories. Indeed, the 2009 and 2019 programs differed in their design, but also in the semantics used. As a result, categories that existed in 2009 disappeared in 2019 and new categories appeared in 2019. To make the comparison possible between the programs, we opted to combine categories according to the definitions proposed in each program. Some definitions, however, did not match precisely and not all the terms were defined. Consequently, category mismatching could have occurred. This concern is reduced by the fact that ambiguous terminology gathered only small numbers. A final shortcoming is that it did not evaluate sex disparity for each session according to the sex of the chair. Indeed, Isbell et al. [32] showed that female chairs respect more sex disparity in the choice of speakers than male chairs. This evaluation of the impact of the sex of the chair on the choice of speaker at the APA annual meetings could be performed in a future study.

## Conclusion

This study showed a positive change in female speaker representation at the APA meetings between 2009 and 2019. It also allowed us to highlight the specific roles, topics and sessions at the meetings where women are still lacking. Highlighting underrepresentation could improve the sex disparity for subsequent meetings [6]. Our results demonstrated that women were more represented at the APA meetings, when compared to the US psychiatry workforce. Even though women make up less than 50% of psychiatrists in the US, it seems important to have a high female representation at the meetings. First, speaking at a meeting increases female

visibility and help younger physicians to find a woman role model in academic psychiatry. Second, the rate of female psychiatrists will probably grow quickly in upcoming years, as around 50% of the residents in psychiatry were women in 2020. We formulated a few hypotheses that could explain this favorable evolution. Additionally, recommendations are now available to help the program committee improve sex disparity at their meetings. Martin [33] listed ten rules to achieve sex equity among speakers at conferences. Interestingly, the first rule is collecting data and the fifth one reporting them, as was the purpose of this paper. Schroeder et al. listed some advice to promote gender equity in meetings, including making gender disaggregated data available [34].

## Supporting information

**S1 Table. Comparison of the proportions of male and female speakers in different session types at the APA 2009 and 2019 annual meetings.**
(DOCX)

**S2 Table. Comparison of the proportion of men and women by role at the APA 2009 and 2019 APA annual meetings.**
(DOCX)

**S3 Table. Comparison of the proportion of men and women among speakers by topic at the 2009 and 2019 APA annual meetings.**
(DOCX)

**S1 Data.**
(XLSX)

## Acknowledgments

The authors thank the university hospital of Tours (CHRU de Tours) for the English editing of this manuscript. They also acknowledge the work of the association "Donner des ELLES à la Santé" for gender equality and women's rights In the field of health.

## Author Contributions

**Conceptualization:** Coraline Hingray, Wissam El-Hage.

**Data curation:** Wayne-Corentin Lambert, Stéphane Sanchez.

**Formal analysis:** Sabrina Sebbane, Sophie Bailly, Wayne-Corentin Lambert, Stéphane Sanchez.

**Methodology:** Coraline Hingray, Wissam El-Hage.

**Resources:** Sabrina Sebbane, Sophie Bailly, Stéphane Sanchez, Wissam El-Hage.

**Software:** Wayne-Corentin Lambert, Stéphane Sanchez.

**Supervision:** Wayne-Corentin Lambert, Coraline Hingray, Wissam El-Hage.

**Validation:** Sabrina Sebbane, Sophie Bailly, Coraline Hingray, Wissam El-Hage.

**Writing – original draft:** Sabrina Sebbane, Sophie Bailly, Coraline Hingray, Wissam El-Hage.

**Writing – review & editing:** Sabrina Sebbane, Sophie Bailly, Wayne-Corentin Lambert, Stéphane Sanchez, Coraline Hingray, Wissam El-Hage.

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
