## [Decision Letter · Decision Letter 0]

22 Apr 2021

PONE-D-20-40415

Changes in sex equity at American Psychiatric Association annual meetings over 10 years (between 2009 and 2019)

PLOS ONE

Dear Dr. Wissam El-Hage, 

Thank you for submitting your manuscript to PLOS ONE. After careful consideration, we feel that it has merit but does not fully meet PLOS ONE’s publication criteria as it currently stands. Therefore, we invite you to submit a revised version of the manuscript that addresses the points raised during the review process.

We do need to highlight whether there is an existent sex disparity or not. Hence, this I think that topic is relevant in terms of academia. However, the topic does not represent the paper well.  I would not use the word 'equity' here. The introduction could have more updated references relating to sex differences in academia. There are several recent papers published on this topic. The methodology is weak. The authors should use hierrchial regression technique in their statistical analysis. That would give more insight into the results once partial pooling is used. At present, frequencies and chi square analysis have been applied in the analysis. This analysis does not do justice to the objective of the paper. The discussion could be made stronger by using updated references.

Once the authors have incorporated the changes suggested by the editor and the reviewers, we would be happy to review the paper again. Thank you.

We look forward to receiving your revised manuscript.

Kind regards,

Sabeena Jalal, MBBS, MSc, MSc, SM

Academic Editor

PLOS ONE

Journal Requirements:

Please improve statistical reporting and refer to p-values as "p<.001" instead of "p<.0001". Our statistical reporting guidelines are available at https://journals.plos.org/plosone/s/submission-guidelines#loc-statistical-reporting

Please include your tables as part of your main manuscript and remove the individual files. Please note that supplementary tables (should remain/ be uploaded) as separate "supporting information" files

Thank you for stating the following in the Competing Interests section:

CH reports personal fees from EISAI, Janssen, Lundbeck, Otsuka and UCB. WEH reports personal fees from EISAI, Janssen, Lundbeck, Otsuka, UCB, Roche and Chugai.

Please amend your list of authors on the manuscript to ensure that each author is linked to an affiliation. Authors’ affiliations should reflect the institution where the work was done (if authors moved subsequently, you can also list the new affiliation stating “current affiliation:….” as necessary).

Please ensure that you refer to Figure 1-2 in your text as, if accepted, production will need this reference to link the reader to the figure.

7. Please include captions for your Supporting Information files at the end of your manuscript, and update any in-text citations to match accordingly. Please see our Supporting Information guidelines for more information: http://journals.plos.org/plosone/s/supporting-information

Reviewers' comments:

Reviewer's Responses to Questions

**Comments to the Author**

1. Is the manuscript technically sound, and do the data support the conclusions?

Reviewer #1: Yes

Reviewer #2: Partly

2. Has the statistical analysis been performed appropriately and rigorously? 

Reviewer #1: Yes

Reviewer #2: No

3. Have the authors made all data underlying the findings in their manuscript fully available?

Reviewer #1: Yes

Reviewer #2: Yes

4. Is the manuscript presented in an intelligible fashion and written in standard English?

Reviewer #1: Yes

Reviewer #2: Yes

5. Review Comments to the Author

Reviewer #1: This study addressed an interesting and nice topic, and especially for the first time in psychiatry. It was appropriately planned and analyzed as well as was written in a well-fashioned manner. The aim of the study, methodology, and conclusions were perfectly synchronized.

Minor suggestion:

The authors mentioned, “National demographic data were obtained from the website of the Association of American Medical Colleges for 2007 and 2017 as data for 2009 and 2019 were not available”. It represents that, these were the latest data, that is data for 2008 and 2018 were also not available. I would like to suggest adding this to the statement.

Reviewer #2: Comments

1. Is the title is a current public health issue? NO

2. Is the objective is clear and measurable? NO

3. Is the method clear and explained well? NO

3.1. Who is your study population?

3.2. What is your study design?

3.3. What is sampling technique?

3.4. What is sample size?

3.5. What about exclusion and inclusion criteria?

3.6. What is dependent variable?

4. Is your data is primarily or secondary? If primarily, how you collect through ten years? Or if secondary data, explain it also.

5. The recommendation needs major revision. Based on specific, measurable, time bound and reasonable.

6. PLOS authors have the option to publish the peer review history of their article (what does this mean?). If published, this will include your full peer review and any attached files.

Reviewer #1: **Yes: **Panchanan Acharjee

Reviewer #2: No

---

## [Author Response · Author response to Decision Letter 0]

5 May 2021

We do need to highlight whether there is an existent sex disparity or not. Hence, this I think that topic is relevant in terms of academia. 

• Response: Thank you for the useful comments to enhance the quality of the paper. 

However, the topic does not represent the paper well. I would not use the word 'equity' here. 

• Response: We understand that sex disparity is a relevant topic. As suggested by the reviewer, we changed the word ‘equity’ in the text, and we replaced it in most cases by the word ‘disparity’. 

We also changed the title of the manuscript as follows: “Representation of women at American Psychiatric Association annual meetings over 10 years (between 2009 and 2019).” Indeed, this title is consistent with the primary objective of our work that was to evaluate the evolution of female speakers representation at APA annual meetings over ten years. In our paper, we highlighted an overall increase of female speakers in 2019 compared to 2009. We also showed that women were still underrepresented in leadership roles and honorific sessions in 2019. 

We also changed the wording where appropriate in the manuscript, replacing the word ‘equity’ in most cases by the word ‘disparity’. 

The introduction could have more updated references relating to sex differences in academia. There are several recent papers published on this topic. 

• Response: As suggested, we updated our references. We added more recent references in the Introduction and Discussion sections as suggested. 

We added the following references: 

15. Arora A, Kaur Y, Dossa F, Nisenbaum R, Little D, Baxter NN. Proportion of Female Speakers at Academic Medical Conferences Across Multiple Specialties and Regions. JAMA 2020;3(9):e2018127. 

16. Cohen-Rosenblum AR, Bernstein JA, Cipriano CA. Gender Representation in Speaking Roles at the American Association of Hip and Knee Surgeons Annual Meeting: 2012-2019. The Journal of Arthroplasty. 2021;S0883540321000383. 

17. Harris KT, Clifton MM, Matlaga BR, Koo K. Gender Representation Among Plenary Panel Speakers at the American Urological Association Annual Meeting. Urology. 2021;150:54‑8. 

32. Schroeder E, Rochford C, Voss M, Gabrysch S. Beyond representation: women at global health conferences. The Lancet. 2019;393:1200‑1.

The methodology is weak. The authors should use hierarchical regression technique in their statistical analysis. That would give more insight into the results once partial pooling is used. At present, frequencies and chi square analysis have been applied in the analysis. This analysis does not do justice to the objective of the paper. The discussion could be made stronger by using updated references.

• Response: We followed your recommendations. As requested, we performed new analyses. 

We added the following to the Statistics section: “We performed a multivariable analysis by logistic regression, evaluating the probability of female participation according to role, type of session, topic and APA year (2019 vs 2009). The descriptive analysis was conducted with SPSS (IBM, Inc., Armonk, NY). The multivariable analysis was conducted using R version 4.0.2 (www.R-project.org). All the tests used were two-tailed. Statistical significance was defined as p-value<0.05.” 

We added the following to the Results section: “Results of the multivariable analysis are presented in Table 2. The probability of female participation increased between 2009 and 2019, with an Odds Ratio of 1.580 (95%CI 1.351-1.850, p<.001). The Area Under the Curve (AUC) was 0.603. No calibration problems were found, the Hosmer-Lemeshow Goodness-of-fit test was not significant (p=0.19).”

We added the Table 2: Multivariable analysis by logistic regression modelling the probability of female participation according to APA session. 

• Response: We made the appropriate changes according to the guidelines.

2. Please improve statistical reporting and refer to p-values as "p<.001" instead of "p<.0001". Our statistical reporting guidelines are available at https://journals.plos.org/plosone/s/submission-guidelines#loc-statistical-reporting

• Response: We made the appropriate changes in statistical reporting as required.

3. Please include your tables as part of your main manuscript and remove the individual files. Please note that supplementary tables (should remain/ be uploaded) as separate "supporting information" files

• Response: As requested, we included the Tables in the main manuscript. We also added “supporting information” files as required.

4. Thank you for stating the following in the Competing Interests section: CH reports personal fees from EISAI, Janssen, Lundbeck, Otsuka and UCB. WEH reports personal fees from EISAI, Janssen, Lundbeck, Otsuka, UCB, Roche and Chugai.

• Response: As requested, we added a Competing Interests section reporting the following: “CH reports personal fees from EISAI, Janssen, Lundbeck, Otsuka and UCB. WEH reports personal fees from Air Liquide, EISAI, Janssen, Lundbeck, Otsuka, UCB, Roche and Chugai. This does not alter our adherence to PLOS ONE policies on sharing data and materials.”

We confirm that the reported competing interests do not alter our adherence to PLOS ONE policies on sharing data and materials. We also removed the competing interests section from the our manuscript. We included our updated Competing Interests statement in your cover letter. 

5. Please amend your list of authors on the manuscript to ensure that each author is linked to an affiliation. Authors’ affiliations should reflect the institution where the work was done (if authors moved subsequently, you can also list the new affiliation stating “current affiliation:….” as necessary).

• Response: We made the appropriate changes as required.

6. Please ensure that you refer to Figure 1-2 in your text as, if accepted, production will need this reference to link the reader to the Figure.

• Response: We added the references for the Figures in the text. We also changed the way we presented the Figures in line with PLOS ONE's style requirements. 

• Response: We added captions for supporting information files at the end of the manuscript. We also added a separate file.

Review Comments to the Author

Please use the space provided to explain your answers to the questions above. You may also include additional comments for the author, including concerns about dual publication, research ethics, or publication ethics. (Please upload your review as an attachment if it exceeds 20,000 characters). 

Reviewer #1:

This study addressed an interesting and nice topic, and especially for the first time in psychiatry. It was appropriately planned and analyzed as well as was written in a well-fashioned manner. The aim of the study, methodology, and conclusions were perfectly synchronized.

Minor suggestion: The authors mentioned, “National demographic data were obtained from the website of the Association of American Medical Colleges for 2007 and 2017 as data for 2009 and 2019 were not available”. It represents that, these were the latest data, that is data for 2008 and 2018 were also not available. I would like to suggest adding this to the statement.

• Response: Many thanks for the proposition. As proposed, we added the following sentence in the manuscript: “Data for 2008 and 2018 were also not available.” 

Reviewer #2: Comments

1. Is the title is a current public health issue? NO

• Response: Our work addressed the evolution of the representation of women in the most attended annual psychiatry meeting. It also questioned women actual place in academic psychiatry in general. Thus, this work is indeed not at public health issue but , we believe, an important society issue. Hence, we think that topic is relevant in terms of academia. 

2. Is the objective is clear and measurable? NO

• The goal of this work was to explore the evolution of women representation among speakers of the APA annual meetings over 10 years. To measure the evolution of women speakers, we analyzed the evolution of the proportion of women speakers at the APA annual meeting in 2009 and 2019. In order to explore this evolution, we compared the sex distribution of the speakers at the APA meeting in 2019 to the sex distribution of the speakers at the APA meeting in 2019. A variation of the sex distribution was highlighted with a p value< 0.05. 

3. Is the method clear and explained well? NO

• Response: We used the programs of the 2009 and 2019 APA annual meetings to extract the data. Variables collected for each speaker were: sex, role during their intervention, type of the session they took part in and topic of the session.

3.1. Who is your study population? 

• Response: Our study population consisted of the speakers at the 2009 and 2019 APA annual meetings.

3.2. What is your study design? 

• Response: Our study was designed using a retrospective, cross-sectional methodology. We added that in our manuscript.

3.3. What is sampling technique? 

• Response: No sampling was performed. We collected the data for all the speakers of the 2009 and 2019 APA annual meetings. 

3.4. What is sample size? 

• Response: Our study population was of 1093 and 1762 speakers at the 2009 and 2019 APA meetings respectively. We mentioned it in the results. 

3.5. What about exclusion and inclusion criteria?

• Response: Speakers at the APA 2009 or/and 2019 meeting were included in this work. Speakers who participated only in poster presentations were excluded. Sessions which did not exist in 2009 and 2019 were not taken into consideration as well as the speakers who took part in it. 

3.6. What is dependent variable?

• Response: Not relevant here. 

4. Is your data is primarily or secondary? If primarily, how you collect through ten years? Or if secondary data, explain it also.

• Response: We did not collect data through ten years. We collected the data of 2009 and 2019 APA annual meeting programs.

In the programs, the sex of the speakers was not mentioned. We identified the sex of the speakers using the speaker’s name mentioned in the program combined with an internet search. From the meetings programs, we then extracted the type of sessions in which the speaker presented, the speaker’s role in the session and the topic of the session. All these data were compiled in an excel file. To avoid any mistake, the data was reviewed by two of the authors.

5. The recommendation needs major revision. Based on specific, measurable, time bound and reasonable. 

• Response: Thank you for the useful comments to enhance the quality of the paper.

---

## [Decision Letter · Decision Letter 1]

21 Sep 2021

PONE-D-20-40415R1Representation of women at American Psychiatric Association annual meetings over 10 years (between 2009 and 2019)PLOS ONE

Dear Dr. El-Hage,

Thank you for submitting your manuscript to PLOS ONE. After careful consideration, we feel that it has merit but does not fully meet PLOS ONE’s publication criteria as it currently stands. Therefore, we invite you to submit a revised version of the manuscript that addresses the points raised during the review process.

We look forward to receiving your revised manuscript.

Kind regards,

Marc Potenza

Academic Editor

PLOS ONE

Reviewers' comments:

Reviewer's Responses to Questions

**Comments to the Author**

1. If the authors have adequately addressed your comments raised in a previous round of review and you feel that this manuscript is now acceptable for publication, you may indicate that here to bypass the “Comments to the Author” section, enter your conflict of interest statement in the “Confidential to Editor” section, and submit your "Accept" recommendation.

Reviewer #1: All comments have been addressed

Reviewer #3: (No Response)

Reviewer #4: (No Response)

2. Is the manuscript technically sound, and do the data support the conclusions?

Reviewer #1: Partly

Reviewer #3: No

Reviewer #4: Yes

3. Has the statistical analysis been performed appropriately and rigorously? 

Reviewer #1: I Don't Know

Reviewer #3: No

Reviewer #4: Yes

4. Have the authors made all data underlying the findings in their manuscript fully available?

Reviewer #1: Yes

Reviewer #3: (No Response)

Reviewer #4: Yes

5. Is the manuscript presented in an intelligible fashion and written in standard English?

Reviewer #1: Yes

Reviewer #3: No

Reviewer #4: Yes

6. Review Comments to the Author

Reviewer #1: I had some minor revisions, all were addressed. The study can be published, on the merit of their specific presentation on the change in participation of female; if queries by other reviewers are addressed properly.

Reviewer #3: Referee Report

PLOS ONE, PONE-D-20-40415R1

Representation of women at American Psychiatric Association annual meetings over 10 years (between 2009 and 2019)

A. Summary

This paper studies the change in women’s representation in academic Psychiatry, which is an important and timely topic, by comparing the female share of women in the American Psychiatric Association annual meetings in 2009 versus 2019.

B. Overall takeaway

This paper finds that the percentage of female speakers was larger in 2019 than 2009, which implies that the representation of women at the APA annual meetings has increased during the period.

C. Major comments

1. Statistical method and interpretation

As the authors responded to reviewer #2, the population of this study is “the speakers at the 2009 and 2019 APA annual meetings,” and the data of this study are not sample data, but population data: “We collected the data for all the speakers of the 2009 and 2019 APA annual meetings.” Here, the parameters of interest are the percentages of women of all the speakers in the meetings, which can be found by simply calculating the percentage using the population data the authors have. In other words, the parameters of interest are known, so it is inappropriate to perform any statistical estimation (i.e. z-test, Chi2 analysis, p-value, etc.), which is the process to use sample estimates to approximate the value of unknown population parameters. See similar previous papers (e.g. Gerull et al. 2020, Sleeman et al. 2019, and May and Dimand 2019) which simply report the number/percentage of women without any tests and perform statistical tests only when they need to estimate some unknown parameters.

In the result section, the authors repetitively report estimates, p-values, and statistical significance of the estimates. For the reasons I mentioned above, however, I am neither persuaded that these statistical analyses in this study are well performed, nor that the interpretation and discussion of the results are presented in an appropriate/intelligible fashion.

2. Logistic regression

The authors mechanically report the result of the logistic regression without any interpretation. I cannot find any consideration as to why the explanatory variables are chosen or what the estimates mean. The authors added this analysis responding to a previous reviewer who suggested to use a hierarchical regression technique, but the logistic regression was not performed as such.

3. Suggestion

This paper would be more substantial if the authors went beyond the current research question: whether the percentage of women in the APA meetings increased between 2009 and 2019. The question would be easily answered by simply calculating the percentage of women in the data without any statistical inferences. A possible question which can be further studied with the data is which factor explains the increase in the women’s representation (this is the question where statistical estimation is really needed). Although the authors suggest some possible explanations in the discussion section, they are weakly supported by the data. For example, an interesting question left for future research is “the impact of the sex of the chair on the choice of speaker at the APA annual meetings,” as suggested by the authors in “Strengths and limitations.” If the data provide the information on chairs of each APA meeting session, the authors would be able to regress the percentage of women in a session on the gender of the chair of the session controlling for other explanatory variables.

D. Minor comments

1. Consistency with the national demographic data

The authors could provide more empirical analyses, especially in terms of the limitations of this paper mentioned in the subsection “Strengths and limitations.” For example, one of the limits is that the speakers of the APA meetings are not necessarily U.S. researchers; they also consist of researchers from other countries. The authors could reduce this concern by collecting data on the speakers’ institutions (as the authors collected the gender of the speakers) to determine whether the speakers working within or without the U.S.

2. Referring to figures

Although a previous reviewer suggested to “ensure that you refer to Figure 1-2 in your text,” such figures have not been explicitly referred to by the text. The authors could mention figures 1-3 somewhere in the subsections “Roles,” “Topics,” or “Sessions” of the Results section.

E. Citations

Katherine M. Gerull, Brandon Malik Wahba, Laurel M. Goldin, Jared McAllister, Andrew Wright, Amalia Cochran, Arghavan Salles. Representation of women in speaking roles at surgical conferences. The American Journal of Surgery. 2020;220(1):20-26.

Katherine E. Sleeman, Jonathan Koffman, Irene J. Higginson. Leaky pipeline, gender bias, self-selection or all three? A quantitative analysis of gender balance at an international palliative care research conference. BMJ Supportive & Palliative Care. 2019;9:146-148.

Ann Mari May, Robert W. Dimand. Women in the Early Years of the American Economic Association: A Membership beyond the Professoriate Per Se. History of Political Economy. 2019;51(4):671–702.

Reviewer #4: I applaud the effort of the authors to study the trends and gender differences in APA conferences over the years. The inclusion of women at conferences is similar to that of workforce. However, still there is no parity in the workforce which is likely impact of slower growth.

Abstract:

It will be advisable to add how sex was determined in the study in methods.

Introduction:

Well written.

There are new acronyms being used such as WIM, WIC. It will be good to use WIP: women in psychiatry as a new acronym and use it throughout the manuscript.

APA is not defined and used directly.

Methods:

The design is intuitive. However, would need to clarify and improve a few things.

How and who made the decision regarding adding unmentioned 2009 topics to enter into a different group in 2019.

As mentioned some authors were included in multiple sessions, how about comparing the repetition of male vs female authors. As in the past similar studies, women have fewer repetition compared to men.

https://www.acgme.org/About-Us/Publications-and-Resources/Graduate-Medical-Education-Data-Resource-Book/

Results:

The above website provided updated data up to 2019 for women pshyciatry fellows, those could be mentioned rather than 2007 and 2017.

“Compared to 2009, the proportion of female chairs increased by 12% in 2019, though not significantly (42% vs 47%; p=0.70).”

There seems to be an increase of 5%.

Also, it will be good to give absolute number first and then percentages in brackets to show the numerical increase as well and in compliance with scientific documentation.

It will be also good for the results section to be direct rather than negating it.

Eg.:

“Compared to 2009, the proportion of female chairs

increased by 12% in 2019, though not significantly (42% vs 47%; p=0.70).

Can be modified to

The proportion of female chairs remained similar from xxx(42%) in 2009 to yyy(47%) in 2020.

Or

There was no significant trend in ….

This can be implied throughput and the word count will decrease.

Eg: The percentage of female speakers in addiction psychiatry decreased dropped by 21%,

Remove either one of the bolded as they are synonymous

Discussion:

Starts with again mentioning about methods and rationale which can be cut short for the first paragraph by removing first 2 lines.

The discussion seems to be very well written and compared with all specialties.

It will be also good to compare with cardiology and women leadership as well in the discussions section and consider citing the following paper.

https://www.ahajournals.org/doi/full/10.1161/CIRCOUTCOMES.120.007578

https://www.sciencedirect.com/science/article/pii/S2589790X21000986

https://www.ahajournals.org/doi/full/10.1161/CIRCULATIONAHA.119.044693

7. PLOS authors have the option to publish the peer review history of their article (what does this mean?). If published, this will include your full peer review and any attached files.

Reviewer #1: No

Reviewer #3: No

Reviewer #4: No

---

## [Author Response · Author response to Decision Letter 1]

1 Nov 2021

Reviewers' comments:

Reviewer's Responses to Questions

Comments to the Author

1. If the authors have adequately addressed your comments raised in a previous round of review and you feel that this manuscript is now acceptable for publication, you may indicate that here to bypass the “Comments to the Author” section, enter your conflict of interest statement in the “Confidential to Editor” section, and submit your "Accept" recommendation.

Reviewer #1: All comments have been addressed

Reviewer #3: (No Response)

Reviewer #4: (No Response)

2. Is the manuscript technically sound, and do the data support the conclusions?

Reviewer #1: Partly

Reviewer #3: No

Reviewer #4: Yes

3. Has the statistical analysis been performed appropriately and rigorously?

Reviewer #1: I Don't Know

Reviewer #3: No

Reviewer #4: Yes

4. Have the authors made all data underlying the findings in their manuscript fully available?

Reviewer #1: Yes

Reviewer #3: (No Response)

Reviewer #4: Yes

5. Is the manuscript presented in an intelligible fashion and written in standard English?

Reviewer #1: Yes

Reviewer #3: No

Reviewer #4: Yes

6. Review Comments to the Author

Reviewer #1:

I had some minor revisions, all were addressed. The study can be published, on the merit of their specific presentation on the change in participation of female; if queries by other reviewers are addressed properly.

- Response: We are grateful to Reviewer #1 for the comments that helped us improving the manuscript. 

--- 

Reviewer #3: Referee Report

PLOS ONE, PONE-D-20-40415R1

Representation of women at American Psychiatric Association annual meetings over 10 years (between 2009 and 2019)

A. Summary

This paper studies the change in women’s representation in academic Psychiatry, which is an important and timely topic, by comparing the female share of women in the American Psychiatric Association annual meetings in 2009 versus 2019.

B. Overall takeaway

This paper finds that the percentage of female speakers was larger in 2019 than 2009, which implies that the representation of women at the APA annual meetings has increased during the period. 

C. Major comments

1. Statistical method and interpretation

As the authors responded to reviewer #2, the population of this study is “the speakers at the 2009 and 2019 APA annual meetings,” and the data of this study are not sample data, but population data: “We collected the data for all the speakers of the 2009 and 2019 APA annual meetings.” Here, the parameters of interest are the percentages of women of all the speakers in the meetings, which can be found by simply calculating the percentage using the population data the authors have. In other words, the parameters of interest are known, so it is inappropriate to perform any statistical estimation (i.e. z-test, Chi2 analysis, p-value, etc.), which is the process to use sample estimates to approximate the value of unknown population parameters. See similar previous papers (e.g. Gerull et al. 2020, Sleeman et al. 2019, and May and Dimand 2019) which simply report the number/percentage of women without any tests and perform statistical tests only when they need to estimate some unknown parameters.

In the result section, the authors repetitively report estimates, p-values, and statistical significance of the estimates. For the reasons I mentioned above, however, I am neither persuaded that these statistical analyses in this study are well performed, nor that the interpretation and discussion of the results are presented in an appropriate/intelligible fashion.

- Response: As suggested, we removed the estimates of statistical significance. Descriptive statistics were recalculated. We changed the presentation of our Results section in order to make it easier to read. 

2. Logistic regression

The authors mechanically report the result of the logistic regression without any interpretation. I cannot find any consideration as to why the explanatory variables are chosen or what the estimates mean. The authors added this analysis responding to a previous reviewer who suggested to use a hierarchical regression technique, but the logistic regression was not performed as such.

- Response: We added an explanation of the choice of variables included in the model. “The variables to be included in the logistic regression model were selected to reflect the main differences in congress structure between 2009 and 2019.”

We provided an explanation of the information added by the regression model in the Discussion. “Overall, women were more represented in the 2019 APA conference, and the multivariable analysis shows that this increase was independent of the differences in the repartition of topics, roles, or type of sessions observed in 2009 and 2019.”

We performed a simple (non-hierarchical) logistic regression model because the structure of the data available to us was not well suited for multilevel modelling. This is because although we could ascertain the topic of a presentation for any participant, in our final database we cannot distinguish which participants attended the same presentation together (there was no unique session ID in the final database). Therefore, we cannot cluster participants by individual presentations.

3. Suggestion

This paper would be more substantial if the authors went beyond the current research question: whether the percentage of women in the APA meetings increased between 2009 and 2019. The question would be easily answered by simply calculating the percentage of women in the data without any statistical inferences. A possible question which can be further studied with the data is which factor explains the increase in the women’s representation (this is the question where statistical estimation is really needed). 

- Response: The topics of the sessions attended are available in our data, however the names of the sessions themselves have been removed. Therefore, as there can be multiple sessions with the same topic on the same day, we cannot link individual participants with the chair of their session.

Although the authors suggest some possible explanations in the discussion section, they are weakly supported by the data.

- Response: We removed some parts of the Discussion that were not directly linked to our results. 

For example, an interesting question left for future research is “the impact of the sex of the chair on the choice of speaker at the APA annual meetings,” as suggested by the authors in “Strengths and limitations.” If the data provide the information on chairs of each APA meeting session, the authors would be able to regress the percentage of women in a session on the gender of the chair of the session controlling for other explanatory variables.

- Response: As our database does not have the names of each unique session, unfortunately we are not able to conduct this analysis.

D. Minor comments

1. Consistency with the national demographic data

The authors could provide more empirical analyses, especially in terms of the limitations of this paper mentioned in the subsection “Strengths and limitations.” For example, one of the limits is that the speakers of the APA meetings are not necessarily U.S. researchers; they also consist of researchers from other countries. The authors could reduce this concern by collecting data on the speakers’ institutions (as the authors collected the gender of the speakers) to determine whether the speakers working within or without the U.S.

- Response: Unfortunately, this information is not available in our database.

2. Referring to figures

Although a previous reviewer suggested to “ensure that you refer to Figure 1-2 in your text,” such figures have not been explicitly referred to by the text. The authors could mention figures 1-3 somewhere in the subsections “Roles,” “Topics,” or “Sessions” of the Results section.

- Response: We made the appropriate changes as suggested.

E. Citations

Katherine M. Gerull, Brandon Malik Wahba, Laurel M. Goldin, Jared McAllister, Andrew Wright, Amalia Cochran, Arghavan Salles. Representation of women in speaking roles at surgical conferences. The American Journal of Surgery. 2020;220(1):20-26.

Katherine E. Sleeman, Jonathan Koffman, Irene J. Higginson. Leaky pipeline, gender bias, self-selection or all three? A quantitative analysis of gender balance at an international palliative care research conference. BMJ Supportive & Palliative Care. 2019;9:146-148.

Ann Mari May, Robert W. Dimand. Women in the Early Years of the American Economic Association: A Membership beyond the Professoriate Per Se. History of Political Economy. 2019;51(4):671–702.

- Response: We made the appropriate changes to the references section. 

--- 

Reviewer #4: I applaud the effort of the authors to study the trends and gender differences in APA conferences over the years. The inclusion of women at conferences is similar to that of workforce. However, still there is no parity in the workforce which is likely impact of slower growth.

- Response: We thank the Reviewer #4 for these encouragements. 

Abstract:

It will be advisable to add how sex was determined in the study in methods.

- Response: We changed the Methods section to clarify this issue: “The sex of the speakers was identified using their first and last names, combined with an Internet search, which was necessary to confirm cases for which direct identification was not possible. Sex identification was performed by one author, then verified by a second one to avoid wrong sex assignment.” 

Introduction:

Well written.

- Response: Thank you for this positive comment. 

There are new acronyms being used such as WIM, WIC. It will be good to use WIP: women in psychiatry as a new acronym and use it throughout the manuscript.

- Response: We added the acronym “WIP: women in psychiatry” and used it throughout the revised manuscript.

APA is not defined and used directly.

- Response: We added the definition of APA in the revised manuscript.

Methods:

The design is intuitive. However, would need to clarify and improve a few things.

How and who made the decision regarding adding unmentioned 2009 topics to enter into a different group in 2019. 

- Response: We clarified this point as follows: “As some of the topics mentioned in 2009 did not match those in 2019 and were more complex, the authors decided after collegiate discussion to assign one of the 2019 topics to a part of the 2009 sessions. Sessions with no topic mentioned or no topic corresponding to the list were labeled as “other topics”.”

As mentioned some authors were included in multiple sessions, how about comparing the repetition of male vs female authors. As in the past similar studies, women have fewer repetition compared to men.

- Response: Concerning the repetition, we have available data mentioned in the text: “The number of interventions per speaker decreased for both genders: from 1.26 to 1.14 interventions per speaker for women and from 1.57 to 1.40 interventions per speaker for men.”

However the names of the sessions themselves have been removed. Therefore, as there can be multiple sessions with the same topic on the same day, we cannot link individual participants with the chair of their session.

https://www.acgme.org/About-Us/Publications-and-Resources/Graduate-Medical-Education-Data-Resource-Book/

Results:

The above website provided updated data up to 2019 for women psychiatry fellows, those could be mentioned rather than 2007 and 2017.

- Response: We thank the reviewer #4 for sharing this valuable resource. Accordingly, we changed the data in the text.

Also, it will be good to give absolute number first and then percentages in brackets to show the numerical increase as well and in compliance with scientific documentation.

- Response: We complied to this convention in the revised manuscript. 

It will be also good for the results section to be direct rather than negating it.

E.g.: “Compared to 2009, the proportion of female chairs increased by 12% in 2019, though not significantly (42% vs 47%; p=0.70). 

Can be modified to: “The proportion of female chairs remained similar from xxxx (42%) in 2009 to yyyy (47%) in 2020. “

Or: “There was no significant trend in ….”

This can be implied throughput and the word count will decrease.

- Response: As suggested, we made the appropriate changes in the revised manuscript. 

E.g.: The percentage of female speakers in addiction psychiatry decreased dropped by 21%. Remove either one of the bolded as they are synonymous

- Response: We made the appropriate correction in the revised manuscript. 

Discussion:

Starts with again mentioning about methods and rationale which can be cut short for the first paragraph by removing first 2 lines.

- Response: We removed these lines in the revised manuscript. 

The discussion seems to be very well written and compared with all specialties.

- Response: We thank the reviewer #4 for this appreciation of our work. 

It will be also good to compare with cardiology and women leadership as well in the discussions section and consider citing the following paper.

https://www.ahajournals.org/doi/full/10.1161/CIRCOUTCOMES.120.007578

https://www.sciencedirect.com/science/article/pii/S2589790X21000986

https://www.ahajournals.org/doi/full/10.1161/CIRCULATIONAHA.119.044693

- Response: We thank the reviewer for this resource and cited these papers in the revised manuscript.

---

## [Decision Letter · Decision Letter 2]

24 Nov 2021

Representation of women at American Psychiatric Association annual meetings over 10 years (between 2009 and 2019)

PONE-D-20-40415R2

Dear Dr. El-Hage,

We’re pleased to inform you that your manuscript has been judged scientifically suitable for publication and will be formally accepted for publication once it meets all outstanding technical requirements.

Kind regards,

Marc Potenza

Academic Editor

PLOS ONE

Additional Editor Comments (optional):

Reviewers' comments:

Reviewer's Responses to Questions

**Comments to the Author**

1. If the authors have adequately addressed your comments raised in a previous round of review and you feel that this manuscript is now acceptable for publication, you may indicate that here to bypass the “Comments to the Author” section, enter your conflict of interest statement in the “Confidential to Editor” section, and submit your "Accept" recommendation.

Reviewer #4: All comments have been addressed

2. Is the manuscript technically sound, and do the data support the conclusions?

Reviewer #4: Yes

3. Has the statistical analysis been performed appropriately and rigorously? 

Reviewer #4: Yes

4. Have the authors made all data underlying the findings in their manuscript fully available?

Reviewer #4: Yes

5. Is the manuscript presented in an intelligible fashion and written in standard English?

Reviewer #4: Yes

6. Review Comments to the Author

Reviewer #4: The authors have done a commendable job on addressing all the issues. They are addressing an important issue which is important to address in the current era.

7. PLOS authors have the option to publish the peer review history of their article (what does this mean?). If published, this will include your full peer review and any attached files.

Reviewer #4: No